# Domain-Specific Stimulation of Executive Functioning in Low-Performing Students with a Roma Background: Cognitive Potential of Mathematics

**Iveta Kovalčíková [1], Jochanan Veerbeek [2]** **, Bart Vogelaar [3]** **, Alena Prídavková [4],***, **Ján Ferjenčík [5], Edita Šimčíková [4]** and **Blanka Tomková [4]**

1   Research Center of Cognitive Education, Faculty of Education, University of Presov in Presov, 080 01 Prešov, Slovakia; iveta.kovalcikova@unipo.sk
2   Education and Child Studies, Leiden University, 2333 AK Leiden, The Netherlands; j.veerbeek@fsw.leidenuniv.nl
3   Developmental and Educational Psychology, Leiden University, 2333 AK Leiden, The Netherlands; b.vogelaar@fsw.leidenuniv.nl
4   Department of Mathematic Education, Faculty of Education, University of Presov in Presov, 080 01 Prešov, Slovakia; edita.simcikova@unipo.sk (E.Š.); blanka.tomkova@unipo.sk (B.T.)
5   Department of Psychology, Faculty of Arts, Pavol Jozef Safarik University in Kosice, 040 01 Košice, Slovakia; jan.ferjencik@upjs.sk
*   Correspondence: alena.pridavkova@unipo.sk; Tel.: +421-517470542

**Abstract:** The current study investigated whether a domain-specific intervention of ExeFun-Mat targeting math and executive functions in primary school children with a Roma background would be effective in improving their scholastic performance and executive functioning. ExeFun-Mat is based on the principles of the reciprocal teaching approach, scaffolding and self-questioning. The domain-specific content was divided into modules. Each module consisted of a set of graded tasks. The criteria for the grading and hierarchical organization of the tasks were based on the level of cognitive difficulty and the type of representation. In total, 122 students attending grade four of elementary school took part in the project. The study concerned a pretest-intervention-posttest experimental design with three conditions: the experimental condition, an active, and a passive control group. To assess the children's level of EF, the Delis–Kaplan executive function system test battery was used; to assess children's mathematical achievement, the cognitive abilities test (the numeracy battery), and ZAREKI—a neuropsychological test battery for numerical processing and calculation—were used. The results suggested that both math performance and executive functions improved over time, with no significant differences between the three conditions. An additional correlational analysis indicated that pretest performance was not related to posttest performance for the children in the experimental and active control group.

**Keywords:** executive functioning; domain-specific cognitive stimulation; math; ExeFun-Mat; low-performing student; Roma ethnic group

## 1. Introduction

In many countries, poor mathematics achievement is often seen in students from low-income and ethnic minority backgrounds [1–3]. One of these groups in the Slovak Republic encompasses the Roma. The Roma are "an extraterritorial ethnic group living in the ethnic environment in the form of an intra-differentiated diaspora" [4], who originated in the Indian subcontinent. The oldest-known written reference to the Roma in Europe dates to 1068, and the oldest-known reports of the Roma in Slovakia are from the second half of the 14th century [5]. In this context, it is important to note that the Roma ethnic minority in Slovakia is highly structured and heterogeneous, not only in terms of ethnic

subgroups and dialects (e.g., Rumunger, Olas, Valachrom) but also in terms of their social integration, economic status and education [6].

In the Slovak Republic, levels of mathematical achievement tend to be much lower in Roma children than in the general school population [7,8]. Moreover, failure rates among the Roma are higher than among the general school population, and they are often placed in special education programs [9–11].

In general, performing well in mathematics seems to be related to various external factors, such as the education system, school, teacher, and family, as well as internal factors, such as the child's personality traits, motivation, attitude, and math anxiety [12]. One important internal factor influencing mathematics achievement concerns the child's executive functioning ability (e.g., [13]). Several studies suggest that children with weaker executive functions tend to perform poorly on mathematics, at the preschool age [14], and in primary [15], as well as in secondary, education [16]. One study [17] suggested that children with a Roma background in Slovakia did indeed score significantly lower on executive functions than children with a majority background. However, to date, it has not been researched whether an intervention targeting executive functioning would also be effective for the mathematics performance of children with a Roma background. Therefore, the current study aimed to investigate whether a newly developed intervention stimulating executive functioning, ExeFun-Mat (executive functioning stimulation via mathematics), would be beneficial in improving Roma children's executive functioning and scholastic performance in the field of mathematics.

## 2. The Current Study

Utilizing a pretest-training-posttest design with an experimental condition, and an active control and a passive control condition, the current study aimed to investigate whether the ExeFun-Mat (executive functioning stimulation via mathematics) intervention had an effect on Roma children's executive functioning and educational performance.

The first research question concerned the potential effect of the ExeFun-Mat program on children's executive functioning. It was hypothesized that those children who received the ExeFun-Mat intervention would show more improvement in their executive functions, specifically inhibition, cognitive flexibility, self-regulation, attention control, and planning, than those in the two control conditions [18–21].

The second research question concerned the potential effect of the ExeFun-Mat program on children's mathematical performance. In accordance with previous studies, it was expected that the children who received the ExeFun-Mat program would show more improvement in their mathematics performance than those in the two control conditions [22–25].

The third research question concerned the potential effect of the ExeFun-Mat program on the relationship between children's executive functioning, math abilities and math performance in school. Based on several studies [26–28], it was expected that the relationship between executive functioning and math performance in school would become weaker for children who received the ExeFun-Mat program from pretest to posttest, but not for those in the two control conditions. Similarly, it was explored whether the relationship between children's mathematical abilities and math performance in school would become weaker for children who received the ExeFun-Mat program from pretest to posttest, but not for those in the two control conditions.

### 2.1. Theoretical Framework

2.1.1. The Influence of Executive Functioning on Achievement in Mathematics

The term "executive functioning" refers to the mechanism by which performance is optimized in situations requiring the operation of a number of cognitive processes [29]. The term is generally used to represent an umbrella construct that includes a collection of interrelated functions that are responsible for purposeful, goal-directed, problem-solving behavior [30].

Research on the impact of executive functions in relation to school performance shows that they are a better predictor than IQ scores, mathematical skill, or level of reading literacy (e.g., [31,32]). Specific executive functions found to influence mathematics performance include working memory [33], inhibition [34], and cognitive flexibility [35], as well as the higher-order function, attention [36,37]. Moreover, a meta-analysis by Cragg and Gilmore [18] further showed that skills linked to executive functions, such as monitoring and manipulating information in the mind (working memory), suppressing unwanted stimuli (inhibition), and flexible thinking (cognitive flexibility), played an important role in the development of mathematical knowledge and skills.

### 2.1.2. Executive Function Training

In recent studies, it was found that executive functioning can be strengthened as a consequence of intervention [18,19,21,38,39]. In general, two types of interventions can be distinguished when it comes to executive functioning: domain-general and domain-specific interventions. Several studies indicated that interventions aimed at strengthening executive functioning have a positive effect on children's mathematical performance [23–25]. In a study by Goldin et al. [40] it was further found that a computer-based intervention targeting executive functioning had an equalizing effect on the academic and mathematics achievement of children from lower social-economic backgrounds.

In accordance with this opinion group, the current study sought to investigate whether ExeFun-Mat, a newly developed domain-specific intervention for mathematics, could stimulate math performance and executive functioning in low-performing students with a Roma background.

### *2.2. Materials and Methods*

### 2.2.1. Participants

The participants were low-performing students from segregated Roma communities in Slovakia, attending grade four of mainstream elementary schools in rural areas of Slovakia. Students were selected if, according to their teachers, they had achieved below-average results on math tests in the three years prior to the intervention (performance was below average, mark three and lower). Three elementary schools were involved in the project, all of which were located in segregated Roma communities, situated on the outskirts of a town or settlement. None of the students in any of the schools spoke Slovakian as their first language. The students' home language was Romani. All the students attending grade four took part in the project—122 students in total, once parental consent for each child's participation in the project had been obtained. The research sample consisted of (very) low achievers in math, many of them with (1), a history of a grade repetition, (2), low conduct of the language of instruction, and (3), an observed low level of motivation. The cognitive ability numerical battery [41] was used at the pretest to objectify mathematical performance, consisting of three subtests, namely, numerical relations, series of numbers, and compilation of equations. These measures offer an overview of the child's basic quantitative concepts, and their ability to see relationships between them. The results showed an average result of 21.33 (SD = 9.4), which represents 80 points of the weighted score and points to the mathematical abilities of the observed group at average and below-average performance levels. In the frequency analysis, up to 60% of students showed a deficit result ($z = -2$). The 122 children were then randomly divided into three groups (the experimental group and two control groups—active and passive). A mixture of equalization and random selection was used, based on the child's characteristics (sex, place of birth/type of settlement, and mathematics grade from pretest data). Following pretest data, children with equal scores were randomly allocated to one of the three condition groups. The division of participants can be found in Table 1.

**Table 1.** Number of children by group and sex.

| Condition | Boy | Girl | Total |
|---|---|---|---|
| experimental | 19 | 21 | 40 |
| (active) control 1 | 19 | 23 | 42 |
| (passive) control 2 | 19 | 21 | 40 |
| total | 57 | 65 | 122 |

Children's executive functions and math abilities were tested twice—before the experiment (pretests) and two weeks after the experiment, to assess the short-term transfer of the experimental effect (posttests). There was a 3.5-month interval between the pretest and posttest. The tests were clinically administered, individually and during lesson-time at the student's school, and took approximately 60 min. Ten trained data collectors (school psychologists) participated in the project.

2.2.2. Design and Procedure

The study utilized a pretest–intervention–posttest experimental design with three conditions [42]: the experimental group (EG), active control group (C1G), and passive control group (C2G). The EG received the original domain-specific intervention program. The intervention consisted of 30 units, and each unit took 45 min. The intervention was conducted in the school during school time, and was administered twice a week. The active control group was given 30 extra hours of mathematics education in addition to the compulsory school curriculum. The teacher, employed at the project school, worked with a regular mathematics textbook. In the active control group, there was no specific stimulation of executive functioning. The C2G (passive control/contrast group, waiting list group) did not perform any additional tasks. The research was designed with respect to the Code of Ethics of the American Educational Research Association, approved by the AERA Council in February 2011 [43]. The project research design was approved by the Ethics Committee of the Faculty of Education, University of Prešov, under the number 2016/4.

2.2.3. Materials

*Executive functioning (EF).* To assess the children's level of EF, the Delis–Kaplan executive function system [44] D-KEFS test battery was used. The D-KEFS test battery was adapted for use with the Slovak population, and the psychometric characteristics were tested and described by [45]. The internal consistency was below 0.70 in the individual indicators of all battery tests, which from a psychometric point of view tends to be considered as the lower limit of "good" reliability. The individual D-KEFS battery indicators showed moderately high correlations with the W-J battery indicators. In this study, the following subtests from the D-KEFS battery were used: (1) D-KEFS trail-making test—a test of attention organization and flexibility in five test conditions, capable of abstracting interference factors of visual searching and motor speed; (2) D-KEFS verbal fluency, which measures the ability to fluently generate verbal responses to letter prompts and categories within 60 s; (3) D-KEFS design fluency test, a test of figural fluency (in three control conditions) in the visual domain; (4) D-KEFS color–word interference test, a version of the Stroop test in four test conditions that measure the ability to inhibit "learned" behavioral responses.

*Mathematical achievement.* To assess children's mathematical achievement, the following instruments were used:

1. Cognitive abilities test (CogAT) [41]—the numeracy battery (quantitative relations, number series, equation building, pictures). The reliability of the Cognitive Abilities Test numeracy battery is 0.85 [46];

2. ZAREKI [47,48]—a neuropsychological test battery for numerical processing and calculation that provides information on deficits in mathematical ability. ZAREKI has 11 subtests containing 59 tasks in total. These map the basic mathematical skills: perceptual, memory, spatial, verbal, operational as well as mathematical reasoning factors. The internal consistency of a ZAREKI test in a normative sample is 0.90, as recorded in [49].

*ExeFun-Mat stimulation program in the mathematical domain.*

The ExeFun-Mat program consisted of an intervention, in which one trained administrator worked with two students. The intervention was based on the principles of the reciprocal teaching approach [50–52], scaffolding and self-questioning. This included focusing on the student's ability to generate questions, clarifying and summarizing the information they have read, moving from being passive observers of learning to active teachers, and becoming involved in the learning experience as peer tutors.

For the content of the intervention program, the Slovak national mathematics curriculum was used as a source. The program consisted of:

1.  Application of tasks and tasks for developing mathematical thinking (sequences, combinatorics, propositional logic);
2.  Numbers and operations involving natural numbers;
3.  Geometry (basic geometric shapes, two-dimensional and three-dimensional orientation).

According to [53], two key content areas for younger children in learning mathematics are numbers (numbers, operations, relationships) and geometry (spatial perception and thinking, measurement). Based on these criteria, 25 items were constructed. For the selection of the items, 25 expert teachers scored the pool of items based on: importance (with regard to national academic standards in Math); difficulty to teach (perceived methodological problems to deliver this part of the curriculum); difficulty to learn (observed difficulties in students when trying to reach the standard in this area). The 10 highest-scoring items were selected.

The domain-specific content was divided into modules. Each module consisted of a set of graded tasks. The criteria for the grading and hierarchical organization of the tasks were based on the level of cognitive difficulty [54], and the type of representation—enactive, iconic and symbolic modes [55]. The way the items were structured is shown in Table 2.

**Table 2.** Level of difficulty—example of item structure.

| | **Level of Difficulty 1** **Three Elements, Enactive Mode** |
|---|---|
| | We built three-color towers of three cubes (red, yellow and blue). Check that all solutions are correct. (Towers built as the illustration shows). |
| Item 1 | 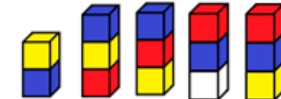 |
| | [ . . . ] 2nd and 3rd level of difficulty |
| Item 4 | Level of difficulty 4 Four elements, iconic mode |
| | Check which buildings in the picture are built according to this rule: The building has three floors and cubes of three colors—red, blue and yellow. 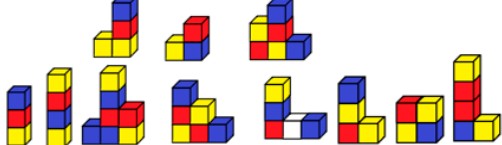 |
| Item 5 | Level of difficulty 5 Three elements, enactive mode |
| | You have a green, yellow and purple square. Draw all the ways you can order them side by side. |
| Item 6 | Level of difficulty 6 Three elements, symbolic mode |

**Table 2.** *Cont.*

| **Level of Difficulty 1**<br>**Three Elements, Enactive Mode** |
|---|
| Three children, Janka, Adam and Beata, went to the cinema. They sat side by side in one row. Write down all the options/combinations for how they could sit down. |
| [ ... ] 7th level of difficulty |

| Item 8 | **Level of difficulty 8**<br>Three elements, symbolic mode |
|---|---|
| | Make all three-digit numbers from digits 4, 7, 5. |
| | [ ... ] |

The intervention was conducted by trained university students studying for a master's degree in teacher training. The students were supervised by the members of the research team. Training of the administrators was over a period of 30 h, and was organized in several recursive cycles while reflecting on the intervention issues and unexpected problems that arose.

*2.3. Results*

2.3.1. Effects of Training on Executive Functions

Initially, the effects of the intervention on children's executive functioning were investigated. A repeated measures MANOVA was conducted, with one "within" factor—time (pretest and posttest), and one "between" factor—condition (experimental condition, control condition 1, and control condition 2). Dependent variables were TMT motor speed time, letter fluency—total correct, category fluency—total correct, switching fluency—total accuracy, design fluency—total correct, and Stroop interference time of the Delis–Kaplan tests. The results of the RM MANOVA are presented in Table 3. The multivariate main-effect of time was significant (Wilks' $\lambda = 0.40$, $F(6,110) = 27.26$, $p < 0.001$, $\eta_p^2 = 0.60$), indicating that overall, all children showed gains on the measurements from pretest to posttest. However, the multivariate interaction effect of time × condition was non-significant (Wilks' $\lambda = 0.89$, $F(12,220) = 1.12$, $p = 0.346$, $\eta_p^2 = 0.06$), indicating that overall, the conditions did not differ regarding progression from pretest to posttest.

**Table 3.** Results of the repeated measures MANOVA for executive functioning.

| | **Wilks' $\lambda$** | **$F$** | **$p$** | **$\eta_p^2$** |
|---|---|---|---|---|
| Multivariate effects | | | | |
| Time | 0.40 | 27.26 | <0.001 | 0.60 |
| Time × Condition | 0.89 | 1.12 | 0.346 | 0.06 |
| Univariate effects (Time) | | | | |
| TMT Motor Speed | | 18.24 | <0.001 | 0.14 |
| Letter Fluency | | 15.13 | <0.001 | 0.12 |
| Category Fluency | | 39.89 | <0.001 | 0.26 |
| Switching Fluency | | 8.25 | 0.005 | 0.07 |
| Design Fluency | | 80.17 | <0.001 | 0.41 |
| Stroop Interference | | 36.45 | <0.001 | 0.24 |

Follow-up univariate analyses (see Table 3) revealed that overall, children showed significant differences from pretest to posttest on most measures, namely TMT motor speed ($p < 0.001$), letter fluency ($p < 0.001$), category fluency ($p < 0.001$), switching fluency ($p = 0.005$), design fluency ($p < 0.001$), and Stroop interference ($p < 0.001$). When inspecting the means and SDs, it can be seen that, overall, children showed a reduction from pretest to posttest on TMT motor speed ($\Delta M = -13.7$) and Stroop interference ($\Delta M = -13.1$), which indicates that children needed less time to complete these tasks. It can also be seen that,

overall, children showed an increase from pretest to posttest on letter fluency ($\Delta M = 1.5$), category fluency ($\Delta M = 2.1$), switching fluency ($\Delta M = 0.7$), and design fluency ($\Delta M = 4.5$).

### 2.3.2. Effects of Training on Math Performance

In order to investigate the effects of the intervention on math abilities, a repeated measures MANOVA was conducted. The RM MANOVA had one "within" factor—time (pretest and posttest), and one between factor—condition (experimental condition, control condition 1, and control condition 2). The dependent variables of the Cogat test were "Quantitative Reasoning Total Correct" and "Inductive Reasoning Condition 3 Total Correct". The dependent variables of arithmetical ability were: arithmetical ability—total, enumeration, counting backward, writing numbers, mental calculation, mental calculation—deduction, reading numbers, number line estimation, magnitude words, perception quantity, context magnitude, problem-solving, and magnitude Arabic numbers (Table 4).

**Table 4.** Means and standard deviations for the measures of executive functioning and math abilities.

| | Pretest | | | Posttest | | |
|---|---|---|---|---|---|---|
| | Experimental M (SD) | Active Control M (SD) | Passive Control M (SD) | Experimental M (SD) | Active Control M (SD) | Passive Control M (SD) |
| **Executive Functions** | | | | | | |
| TMT Motor Speed | 73.15 (30.51) | 74.40 (33.07) | 75.20 (33.19) | 57.82 (25.90) | 64.41 (34.86) | 60.42 (30.50) |
| Letter Fluency | 8.77 (4.07) | 8.18 (4.83) | 6.57 (4.28) | 10.00 (4.65) | 10.08 (4.64) | 7.82 (4.74) |
| Category Fluency | 17.38 (4.07) | 18.31 (4.61) | 14.90 (5.23) | 20.25 (5.36) | 19.77 (4.87) | 16.80 (4.14) |
| Switching Fluency | 6.57 (2.22) | 5.79 (2.46) | 5.62 (1.93) | 7.32 (2.07) | 6.08 (2.36) | 6.22 (2.11) |
| Design Fluency | 3.35 (1.87) | 3.59 (2.01) | 2.28 (1.59) | 4.55 (1.97) | 3.79 (1.98) | 3.22 (2.07) |
| Stroop Interference | 106.18 (23.54) | 109.92 (30.78) | 110.79 (25.75) | 93.70 (22.65) | 95.28 (24.30) | 98.10 (20.56) |
| **Math Abilities** | | | | | | |
| Quantitative Reasoning | 12.08 (4.31) | 11.00 (4.56) | 9.75 (4.72) | 13.82 (4.53) | 12.25 (3.93) | 10.42 (5.03) |
| Inductive Reasoning | 6.92 (4.38) | 7.08 (4.60) | 4.16 (2.83) | 9.27 (4.64) | 7.55 (4.91) | 5.63 (3.20) |
| Arithmetical Ability Total | 5.38 (2.95) | 4.20 (2.51) | 2.61 (1.64) | 6.25 (3.25) | 4.92 (3.15) | 3.21 (2.59) |
| Enumeration | 1.71 (0.57) | 1.77 (0.60) | 1.83 (0.38) | 1.92 (0.27) | 1.89 (0.40) | 1.69 (0.67) |
| Counting Backward | 1.45 (0.80) | 1.58 (0.77) | 1.50 (0.62) | 1.58 (0.79) | 1.56 (0.81) | 1.65 (0.70) |
| Writing Numbers | 9.42 (2.97) | 9.08 (3.33) | 9.41 (3.03) | 8.82 (3.23) | 8.61 (3.42) | 8.37 (3.61) |
| Mental Calculation | 8.45 (3.09) | 8.11 (3.29) | 8.37 (3.57) | 8.82 (3.23) | 8.61 (3.42) | 8.37 (3.61) |
| Mental Calculation Deduction | 5.68 (3.92) | 7.11 (3.86) | 6.44 (3.60) | 6.47 (3.90) | 6.53 (4.02) | 7.28 (4.36) |
| Reading Numbers | 10.16 (2.52) | 10.39 (2.85) | 10.53 (2.24) | 11.00 (2.36) | 10.61 (2.53) | 10.53 (2.35) |
| Number Line Estimation | 5.74 (2.42) | 6.50 (2.88) | 5.25 (2.82) | 7.16 (2.89) | 7.56 (2.49) | 5.88 (2.92) |
| Magnitude Words | 10.84 (4.15) | 12.22 (2.83) | 12.12 (2.14) | 11.95 (3.46) | 12.44 (2.69) | 12.56 (1.81) |
| Perception Quantity | 3.11 (1.52) | 2.94 (1.31) | 1.75 (1.81) | 3.05 (1.29) | 3.28 (2.40) | 2.25 (1.59) |
| Context Magnitude | 5.47 (3.80) | 4.89 (2.81) | 3.75 (2.63) | 6.42 (3.89) | 5.39 (3.48) | 4.25 (2.77) |
| Problem-Solving | 1.74 (1.69) | 1.78 (1.84) | 2.12 (1.61) | 2.34 (2.07) | 2.25 (1.75) | 2.22 (1.34) |
| Magnitude Arabic Numbers | 12.95 (3.34) | 13.11 (3.56) | 12.87 (2.92) | 13.53 (2.52) | 13.17 (3.33) | 13.06 (2.73) |

The results of the RM MANOVA are presented in Table 5. The multivariate main-effect of time was significant (Wilks' $\lambda = 0.57$, $F(15,86) = 4.40$, $p < 0.001$, $\eta_p^2 = 0.43$), indicating that, overall, all children showed gains on the measurements from pretest to posttest. However, the multivariate interaction effect of time x condition was non-significant (Wilks' $\lambda = 0.75$, $F(30,172) = 0.88$, $p = 0.657$, $\eta_p^2 = 0.13$), indicating that, overall, the conditions did not differ regarding progression from pretest to posttest.

**Table 5.** Results of the repeated measures MANOVA for math abilities.

| | Wilks' λ | F | p | $\eta_p^2$ |
|---|---|---|---|---|
| Multivariate effects | | | | |
| Time | 0.57 | 4.40 | <0.001 | 0.43 |
| Time × Condition | 0.75 | 0.88 | 0.657 | 0.13 |
| Univariate effects (Time) | | | | |
| Quantitative Reasoning | | 22.28 | <0.001 | 0.18 |
| Inductive Reasoning | | 3.79 | 0.054 | 0.04 |
| Arithmetical Ability Total | | 30.92 | <0.001 | 0.24 |
| Enumeration | | 5.42 | 0.022 | 0.05 |
| Counting Backward | | 0.98 | 0.325 | 0.01 |
| Writing Numbers | | 7.97 | 0.006 | 0.07 |
| Mental Calculation | | 0.99 | 0.322 | 0.01 |
| Mental Calculation Deduction | | 0.62 | 0.434 | 0.01 |
| Reading Numbers | | 6.10 | 0.015 | 0.06 |
| Number Line Estimation | | 11.83 | 0.001 | 0.11 |
| Magnitude Words | | 4.39 | 0.039 | 0.04 |
| Perception Quantity | | 1.14 | 0.288 | 0.01 |
| Context Magnitude | | 3.74 | 0.056 | 0.04 |
| Problem-Solving | | 4.73 | 0.032 | 0.05 |
| Magnitude Arabic Numbers | | 2.37 | 0.127 | 0.02 |

Follow-up univariate analyses (see Table 5) revealed that, overall, children showed significant differences from pretest to posttest on most measures, namely quantitative reasoning ($p < 0.001$), arithmetical ability—total ($p < 0.001$), enumeration ($p = 0.022$), writing numbers ($p = 0.006$), reading numbers ($p = 0.015$), number line estimation ($p = 0.001$), magnitude words ($p = 0.039$), and problem-solving ($p = 0.032$). However, children showed no significant differences from pretest to posttest on inductive reasoning ($p = 0.054$), counting backward ($p = 0.325$), mental calculation ($p = 0.322$), mental calculation—deduction ($p = 0.434$), perception—quantity ($p = 0.288$), context—magnitude ($p = 0.056$) and magnitude—Arabic numbers ($p = 0.127$). Overall, children showed an increase from pretest to posttest on quantitative reasoning ($\Delta M = 3.4$), arithmetical ability—total ($\Delta M = 5.2$), enumeration ($\Delta M = 0.1$), writing numbers ($\Delta M = 0.6$), reading numbers ($\Delta M = 0.4$), number line estimation ($\Delta M = 1.0$), magnitude words ($\Delta M = 0.6$), and problem-solving ($\Delta M = 0.4$). These findings indicate that the children had more items correct at the posttest compared to the pretest.

### 2.3.3. Effects of Training on Relationship with Math School Results

Correlations were used to test the relationship between children's executive functions and their math results in school, and changes in this relationship as a result of training. TMT motor speed, letter fluency, category fluency, switching fluency, and Stroop interference were used as measures of executive functions. The results are displayed in Table 6. On the pretest, only design fluency ($r = -0.29$), and Stroop interference ($r = 0.23$) were significantly related to school results in math. Posttest measures for executive functions were split by condition, to test whether different patterns of relationships emerged as a result of training. Here, the only significant correlation was found between design fluency and math results for the experimental condition ($r = -0.48$). No other correlations were found between the measures of executive functions on the posttest and math results, for any of the conditions.

**Table 6.** Correlations between pretest and posttest measures and school results on math.

| | Correlation Pretest × School Result | Correlation Posttest × School Result | | |
|---|---|---|---|---|
| | Total (*n* = 102) | Experimental Condition (*n* = 38) | Control Condition 1 (*n* = 35) | Control Condition 2 (*n* = 29) |
| Executive functions | | | | |
| TMT Motor Speed | −0.01 | −0.02 | −0.06 | 0.08 |
| Letter Fluency | −0.02 | −0.08 | −0.27 | 0.05 |
| Category Fluency | −0.07 | −0.17 | −0.17 | 0.02 |
| Switching Fluency | −0.08 | −0.07 | −0.30 | −0.02 |
| Design Fluency | −0.29 ** | −0.48 ** | −0.10 | −0.28 |
| Stroop Interference | −0.23 * | −0.03 | −0.13 | 0.08 |
| Math abilities | | | | |
| Quantitative Reasoning | −0.38 *** | −0.25 | −0.68 *** | −0.16 |
| Inductive Reasoning | −0.26 ** | −0.23 | −0.09 | −0.21 |
| Arithmetical Ability | −0.52 *** | −0.36 * | −0.67 *** | −0.37 * |
| Enumeration | −0.16 | −0.05 | −0.41 * | −0.23 |
| Counting Backward | −0.27 ** | 0.08 | −0.47 ** | −0.19 |
| Writing Numbers | −0.50 *** | −0.26 | −0.63 *** | −0.33 |
| Mental Calculation | −0.55 *** | −0.30 | −0.66 *** | −0.30 |
| Mental Calculation Deduction | −0.38 *** | −0.39 * | −0.48 ** | −0.33 |
| Reading Numbers | −0.41 *** | −0.24 | −0.65 *** | −0.17 |
| Number line Estimation | −0.32 ** | −0.24 | −0.20 | −0.30 |
| Magnitude Words | −0.37 *** | −0.32 | −0.50 ** | −0.35 |
| Perception Quantity | −0.13 | −0.29 | −0.50 ** | −0.04 |
| Context Magnitude | 0.09 | 0.09 | −0.10 | 0.35 |
| Problem-Solving | −0.33 ** | −0.22 | −0.58 *** | −0.04 |
| Magnitude Arabic Numbers | −0.40 *** | −0.31 | −0.55 ** | −0.40 * |

Note: * = $p < 0.05$. ** = $p < 0.01$. *** = $p < 0.001$.

Additionally, correlations were used to test the relationship between children's math abilities and their math results in school, and changes in this relationship as a result of training. Quantitative reasoning, inductive reasoning, arithmetical ability—total, enumeration, counting backward, writing numbers, mental calculation, mental calculation deduction, reading numbers, number line estimation, magnitude words, perception quantity, context magnitude, problem-solving, and magnitude Arabic numbers were used in the correlation analysis. On the pretest, inductive reasoning and counting backward were weakly related (both $r = -0.27$) to math results, quantitative reasoning, mental calculation deduction, reading numbers, number line estimation, magnitude comparison words, problem-solving, and magnitude Arabic numbers all showed moderate relationships with math results (between $r = -0.32$ and $r = -0.41$). Arithmetical ability, writing numbers, and mental calculations showed strong relationships with math results (ranging from $r = -0.50$ to $r = -0.55$). No relationships were found between enumeration, perceptual quantity and context magnitude, and math results.

Next, the participants were split by condition to investigate the posttest relationships with the same variables. For the experimental conditions, moderate correlations between posttest measures of math ability and math results in school were found for arithmetical ability and mental calculation deduction ($r = -0.36$ and $r = -0.39$ respectively). The other relationships were not significant. For the control 1 condition, moderate correlations (between $r = -0.41$ and $r = -0.50$) were found for enumeration, counting backward, mental calculation deduction, magnitude comparison words, and perceptual quantity. Strong correlations (ranging between $r = -0.56$ and $r = -0.68$) were found for quantitative reasoning, arithmetical ability, writing numbers, mental calculation, reading numbers, problem-solving, and magnitude Arabic numbers. Inductive reasoning, number line estimation, and context magnitude were not related to math results. For the control

2 condition, only arithmetical ability and magnitude Arabic numbers showed significant moderate correlations with math results ($r = -0.37$ and $r = -0.40$ respectively). The other measures were not significantly related to math results. Overall, the control 1 condition appeared to show a similar pattern of relations between math abilities and math results on the posttest, as was found on the pretest. For both the experimental and control 2 conditions, considerably fewer and weaker correlations were found on the posttest.

Interestingly, and in contrast with other studies [33–37] only a few executive functions were (moderately) correlated with math performance.

## 3. Discussion

The current study sought to investigate whether ExeFun-Mat, a newly developed domain-specific intervention for mathematics, could stimulate the math performance and executive functioning of low-performing students with a Roma background. In accordance with Clements et al. [56], it was considered that high-quality mathematics education may have the dual benefit of, on the one hand, teaching math and, on the other hand, facilitating executive function stimulation. Regarding the effect of the ExeFun-Mat program on executive functioning, it was found that executive functions improved over time, but there were no differences between conditions. These findings indicated that in all groups of children, both for those who received the ExeFun-Mat program, and those in the control conditions, executive functions seemed to improve over time, but the experimental condition did not bring about significant improvement. This finding was not in line with studies conducted by [57–61], who studied the potential influence of math domain-specific experimental interventions targeting executive functioning in students, and found a positive effect of their interventions on the participants' executive functioning.

Potential explanations for this finding could lie in Jacob and Parkinson's [62] observation that, although the literature demonstrates a strong correlation between executive function and achievement, the two may not be causally related. In the current study, it was even found that most executive functions were not, and some only weakly, related to mathematical performance in school. This finding was unexpected; however, it corresponds with previous findings in children from special populations, such as those with learning problems, severe arithmetic or language difficulties [63,64], with math anxiety problems [65,66] or children from disadvantaged backgrounds [67]. Our findings are in line with Blakey et al. [68], who found that executive functions mediated the relationship between socioeconomic status and mathematical skills. Children improved over the training, but this did not transfer to untrained executive functions or mathematics. Executive functions may explain socioeconomic attainment gaps, but cognitive training directly targeting executive functions is not an effective way to narrow this gap. Executive functions could simply be a proxy for other background characteristics of the particular child, such as socioeconomic status or a parent's level of education, each of which is highly correlated with both achievement and executive functions. Moreover, researchers have found evidence that both socioeconomic status and family factors are associated with the development of executive functions [69–71], which may have played a role in the current study as well.

With regard to the findings of the current study in the domain of mathematics performance, it was found that all groups of children demonstrated an improvement over time, but no differences were found between the conditions. Although Naglieri and Johnson's [72] findings indicate that students with learning disabilities and mild mental impairments, in particular, could benefit from verbalizing and reflecting on their strategies of arithmetic computation, this was not observed in the current study, in spite of the fact that the ExeFun-Mat program includes extensive verbalizing, thinking aloud and metacognitive monitoring.

A possible explanation regarding this finding concerns the notion that the program was too limited in time to bring about changes in mathematics performance. The research sample consisted of fourth-graders who were (very) low achievers in math, many of

whom were associated with (1), a history of grade repetition, (2), low mastery of the language of instruction, or (3), low levels of motivation. When problems accumulate over several years, a 30-h program probably does not have the capacity to change both executive functioning and math skills. Perhaps the results demonstrated a floor effect, as language issues and deficiencies in basic mathematical concepts could act as barriers to benefiting from metacognitive instruction. More importantly, it cannot be disregarded that the participating children live in conditions of severe poverty. Their living conditions are exacerbated in the winter months, when they suffer from the cold, sleeping problems and malnutrition. In other words, the basic living conditions to facilitate learning may have been absent, nullifying any potential effects a training program might have. It is a known fact that in the case of former travelers and nomads, the school is perceived as a foreign institution, far removed from their needs [4,73]. Considering that the intervention was semi-structured, it was not possible to tailor the program to the individual needs of the participating Roma pupils. Although this warrants further research, it seemed as if these were so significant that the current competencies of the intervention administrators (trained master's students) did not allow for a flexible response to problematic behavioral manifestations related to students' mathematical abilities (e.g., in task administration, "place a square between two triangles", the administrator found that the student did not have a developed spatial concept for "between"). A program that connects more closely to the Roma culture and language might be more suited to remediate such issues. Additionally, future programs might try to actively determine the children's ability level and adaptively provide a program that will connect more closely to children's individual instructional needs, for example, by incorporating a dynamic testing procedure at the start of the program in order to tap into the children's potential for learning and their zone of proximal development [74]. Possible inspiration could be taken from the ideas in the OPEN-MATH project, a conceptual framework for inclusion in mathematics. At the core of the inclusive frameworks lies the dialectics between two facets: social interaction and individual self-determination [75].

The intervention was provided relatively late, as the children already had significant delays in mathematical skills compared to their peers. Devoting extra opportunities to teaching and learning could seem promising, but the timing is very important. As mathematics becomes increasingly difficult in school, the deficits in certain areas become more obvious, and, without early diagnosis and stimulation of the relevant processes, the ability to correct and compensate for these deficits declines. In future research, it seems worthwhile to implement a training intervention such as ExeFun-Mat at a much earlier stage.

The relationship between the different measures and measurement moments showed that the factors related to the performance of the passive control group had not changed from pretest to posttest. In contrast, for both the group that had received the training and the active control group, past performance was no longer related to or predictive of arithmetical performance after training. The fact that the trained children did not seem to differ substantially from the active control group could be explained by a ceiling effect in progress (i.e., the program offered for the active control group was sufficient to "saturate" their instructional needs and provide maximal growth). Taken together, it seems that intervention of some shape or form does "defeat prediction", based on their previous, static results. However, these changes did not lead to improved performance on any of the posttest measures.

In conclusion, the current research found that the ExeFun-Mat program that was implemented among grade-four children from Roma backgrounds did not lead to substantial improvements in their executive functions, nor in their math abilities, over and above those brought about by an alternative educational program. Although previous research has shown that executive functions and math performance are closely related, this finding was not replicated amongst this specific population of children, calling into question whether the relationship between EF and math performance develops in the same

way as for children from other populations, or whether this was related to the instruments and program chosen. Additional research is required to investigate this further. In addition, providing an intervention program did appear to defeat the prediction based on prior math performance only, indicating that children from Roma backgrounds are susceptible to interventions to improve their math abilities. This finding suggests that a different or modified intervention program, perhaps delivered at a different age, might be helpful for these children to fulfill their potential in math. The study establishes an important finding regarding school-based domain-specific intervention, demonstrating that it may not have the desired effect. However, the experiment yielded data informing policy-makers that the absence of compulsory pre-school education for pupils from marginalized communities could cause accumulated problems further on in primary education. Early diagnosis of low-cognitive processing related to school tasks may enable children from marginalized communities to benefit more from classroom teaching. Without specific support from the preschool age onwards, it is likely that these students will not perform adequately, but instead will develop cognitive as well as motivational deficits, which are difficult to address and remediate later. More research needs to be conducted in this field to be able to find support for this tentative conclusion. Regarding the study limitations, we wish to note that social-emotional factors such as student motivation and attitudes were not part of the current study. In future studies, the potential influence of such factors could be further analyzed. From a methodological perspective, the current study supported the notion that the examiners were seen as foreigners from another cultural background. When designing a new intervention for marginalized groups or interpreting intervention outcomes, it is essential that cultural aspects, as well as background variables such as maternal education and socio-economic status [62], are taken into account.

**Author Contributions:** Conceptualization, I.K.; methodology, I.K. and A.P.; software, J.F. and J.V.; validation, I.K. and A.P.; formal analysis, J.F., J.V. and B.V.; investigation, A.P., E.Š. and B.T.; resources, I.K. and A.P.; data curation, J.F., J.V. and B.V.; writing—original draft preparation, I.K., and A.P.; writing—review and editing, J.F.; visualization, I.K.; supervision, I.K.; project administration, I.K.; funding acquisition, I.K. All authors have read and agreed to the published version of the manuscript.

**Funding:** This research was funded by (1) APVV (Slovak Research Agency of Ministry of Education, under the contract APVV-15-0273); (2) ISPA (Proposals to the International School Psychology Research Initiative) and (3) VEGA (Scientific Grant Agency of Ministry of Education and National Academy of Science under the contract 1/0254/20).

**Institutional Review Board Statement:** The study was conducted according to the guidelines of the Declaration of Helsinki, and approved by the Ethics Committee of the Faculty of Education, University of Prešov under the number 2016/4.

**Informed Consent Statement:** Informed consent was obtained from all subjects involved in the study.

**Data Availability Statement:** The data presented in this study are available on request from the corresponding author. The data are not publicly available due to the confidential information involved.

**Conflicts of Interest:** The authors declare no conflict of interest.

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
