# Peer review of "Domain-Specific Stimulation of Executive Functioning in Low-Performing Students with a Roma Background: Cognitive Potential of Mathematics"

_education, doi:10.3390/educsci11060285_

Round 1

Reviewer 1 Report

The current study used a quasi-experimental design to test whether ExeFun-Mat is able to promote students’ math and executive functions within primary school students with Roma background.

This study is well written and carefully thought of (design). Some minor revisions are still needed.

Abstract can be made clear with the inclusion of the instrument used, ExeFun-Mat should also be mentioned, since it is the major intervention used.

Previous studies on ExeFun-Mat should be provided, its implications should be significant to the current study objectives.

Research question – recommend to clearly provide place after the introduction section

Citation needed Line 79 – previous studies…

Ethical concerns should be clearly stated in the method section

IRB information should be provided

The author should also provide grounds with the use of 3 groups as oppose to the experimental control (two group) design – hence, citation needed – line 141

Are there norm scores for the D-KEFS?

Why was repeated measures MANOVA used instead of MANCOVA – just provide justification or cite previous studies done with similar design – appropriateness of statistical treatment needed

Line 319 – 54 should be Clements et al. or Clements and associates [54]

Line 330 - [60] Jacob and Parkinson’s – should be Jacob and Parkinson’s [60]

Please recheck all the references and in-text citations for consistency

Limitations of the study?

Implications for practitioners? For learners?

I sum, the paper is well designed and minor revisions needed to make it clearer.

Author Response

Response to Reviewer 1 Comments

  • Abstract can be made clear with the inclusion of the instrument used, ExeFun-Mat should also be mentioned, since it is the major intervention used.

The following information were added to the abstract:

ExeFun-Mat is based on the principles of the reciprocal teaching approach, scaffolding and self-questioning. The domain-specific content was divided into modules. Each module consisted of a set of graded tasks. The criteria for the grading and hierarchical organization of the tasks were based on (1) level of cognitive difficulty and (2) type of representation......

To assess the children’s level of EF, the Delis–Kaplan Executive Function System test battery was used; to assess children’s mathematical achievement, Cognitive Abilities Test (the numeracy battery) and ZAREKI – a neuropsychological test battery for numerical processing and calculation were used.

The following key word was added: ExeFun-Mat

  • Previous studies on ExeFun-Mat should be provided, its implications should be significant to the current study objectives.

There are no previous studies on ExeFun-Mat, this program is original one, was developed for this particular project purposes. In the close future more studies are planned.   

  • Research question – recommend to clearly provide place after the introduction section

Research question were transferred after introduction section. Subsequently, the numbering of other parts was adjusted.

  • Citation needed Line 79 – previous studies…

The text „Building on previous research“ was removed from the text. After the change, the text looks like this:

Utilizing a pretest-training-posttest design with an experimental condition, and active control and a passive control condition, the current study aimed to investigate whether the ExeFun-Mat (executive functioning stimulation via mathematics) intervention had an effect on Roma children’s executive functioning and scholastic performance.

  • Ethical concerns should be clearly stated in the method section
  • IRB information should be provided

To the part 2.2.2. – Design and procedure the following text was added:

The research was designed with respect to The Code of Ethics of the American Educational Research Association,  approved by the AERA Council in February 2011. The project research design was approved by the Ethics Committee of the Faculty of Education, University of Prešov under the number 2016/4.

https://www.aera.net/About-AERA/AERA-Rules-Policies/Professional-Ethics

After the change, the text looks like this:

The C2G (passive control/contrast group, waiting list group) did not perform any additional tasks. The research was designed with respect to The Code of Ethics of the American Educational Research Association, approved by the AERA Council in February 2011. The project research design was approved by the Ethics Committee of the Faculty of Education, University of Prešov under the number 2016/4.

There is also mentioned in text: All the students attending Grade 4 took part in the project – 122 students in total once parental consent for the child’s participation in the project had been obtained.

  • The author should also provide grounds with the use of 3 groups as oppose to the experimental control (two group) design – hence, citation needed – line 141

The citation was inserted (Ferjenčík, 2000): Ján Ferjenčík: Úvod do metodológie psychologického výskumu. Introduction to methodology of psychological research. Praha: Portál.

  • Are there norm scores for the D-KEFS?

As mentioned in the paper …”Executive functioning (EF). To assess the children’s level of EF, the Delis–Kaplan Executive Function System [42] (now [44]) D-KEFS test battery was used. The D-KEFS test battery was adapted for use with the Slovak population, and the psychometric characteristics were tested and described by [43] (now [45]).

More about the D-KEFS norms:

42. Delis, D.; Kaplan, E.; Kramer, J. The Delis–Kaplan executive function system; The Psychological Corporation: San Antonio, 2001.

43. Ferjenčík, J.; Bobáková, M.; Kovalčíková, I.; Ropovik, I.; Slavkovská, M. Proces a vybrané výsledky slovenskej adaptácie Delis-Kaplanovej systému exekutívnych funkcií D-KEFS. [Process and selected results of Slovak adaptation of Delis-Kaplan system of executive functions D-KEFS]. Československá Psychologie: Časopis Pro Psychologickou Teorii a Praxi 2014, 58, 543-558.

  • Why was repeated measures MANOVA used instead of MANCOVA – just provide justification or cite previous studies done with similar design – appropriateness of statistical treatment needed

A repeated measures MANOVA was used, because there was no a priori expectation of covariates which should be controlled for in the analysis. As such the repeated measures MANOVA provided the appropriate statistical tool to analyze the general effect of the ExeFun-Mat programme on both math abilities and executive functions, as measured with multiple different measures. (Weinfurt, K. P. (2000). Repeated measures analysis: ANOVA, MANOVA, and HLM. In L. G. Grimm & P. R. Yarnold (Eds.), Reading and understanding MORE multivariate statistics (p. 317–361). American Psychological Association)

  • Line 319 – 54 should be Clements et al. or Clements and associates [54] (now [56])

Added to the text Clements and associates

  • Line 330 - [60] Jacob and Parkinson’s – should be Jacob and Parkinson’s [60] (now [62])

Corrected in the text to Jacob and Parkinson’s [60]

  • Please recheck all the references and in-text citations for consistency

All references were made using bibliography software and it means that part “references” was generated automatically.  

The citation style multidisciplinary – digital publishing was used (on the basis of instruction and recommendation for authors) 

  • Limitations of the study? Implications for practitioners? For learners?

This text was added to Discussion part:

This finding suggests that a different or modified intervention program, perhaps delivered at a different age, might be helpful for these children to unfold their potential in maths. The study establishes an important finding regarding school-based domain specific intervention, demonstrating may not have the desired effect. However, on the other hand, the experiment yielded data informing policy makers that absence of compulsory pre-school education for pupils from marginalized communities could cause accumulated problems further in primary education.  Early diagnosis of low-cognitive processing related to school tasks may enable children from marginalized communities to benefit more from classroom teaching. Without specific support from the preschool age onwards, it is likely that these students do not perform adequately, but instead will develop cognitive as well as motivational deficits, which are difficult to address and remediate later. More research needs to be conducted in this field to be able to find support for this tentative conclusion. Regarding study limitations, we wish to note that social emotional factor such as student motivation and attitudes were not part of the current study. In future studies, the potential influence of such factors could be further analyzed. From a methodological perspective, the current study supported the notion that the examiners were seen as foreigners with another cultural background. When designing a new intervention for marginalised groups or interpreting intervention outcomes, it is essential that cultural aspects as well as background variables, such as maternal education, socio-economic status [62]  are taken into account.

Thank you very much for all comments. They helped us to increase the scientific quality and "readability" of the text.

They were very encouraging.

Reviewer 2 Report

Dear authors,

thank you for the opportunity to read the article. When analyzing the arithmetic means in the pre-test, it seems that differences between the groups may differ on the statistical significance level (eg. Letter Fluency 8.77(4.07) 8.18(4.83) 6.57(4.28).  If the division into groups was made by random selection, how can the differences be explained, if they are statistically significant?

The bibliography contains mostly sources focused on the preschool period and please explain if there is a specific reason for it. 

Very interesting work!

Author Response

Response to Reviewer 2 Comments

  • thank you for the opportunity to read the article. When analyzing the arithmetic means in the pre-test, it seems that differences between the groups may differ on the statistical significance level (eg. Letter Fluency 8.77(4.07) 8.18(4.83) 6.57(4.28).  If the division into groups was made by random selection, how can the differences be explained, if they are statistically significant?

In the paper, as shown in the text above Table 1, the division into groups was carried out as a mix of random selection and equivalence based on gender, residence, school marks and pretests. Based on gender, residence, age, grades and five pretest results, groups of three children were created (each trio consisted of children with very similar - but not identical - parameters: age, gender ... etc.) Subsequently in each trio, one member was randomly selected for the experimental group, the second member for control group 1, and finally the third member for control group 2.

This procedure leads to the creation of maximally similar groups and is undoubtedly a much better solution than just simple randomization. However, both simple random selection and its combination with equalization do not guarantee full equivalence in all indicators. It was simply not possible to compile three groups equivalent in all 21 variables. As a one-way ANOVA showed - our procedure has led to the comparison of groups in the vast majority of variables - but there are also a few where these groups are not equivalent. In short, the answer to the reviewer's question (If the division into groups was made by random selection, how can the differences be explained, if they are statistically significant?): Random selection cannot guarantee that individual groups are equivalent in several dozen variables. By combining random selection and balancing, we achieved that the groups were equivalent in almost all significant variables. We recorded statistically significant differences in only five of the 21 variables presented).

  • The bibliography contains mostly sources focused on the preschool period and please explain if there is a specific reason for it. 

Cca 11 (15%) sources are focused on the preschool period;  

cca 35 sources (nearly 50%) are focused on 7-15 age children  

Research was carried out in the given group: influence of executive functioning on mathematical skills that are important for school success, performance in school; the sample also included children from low income families (14, 36 – now 20). Assessment of mathematical abilities were carried out in different age periods (from preschool age, through the 1st year to the 3rd year of elementary school) - which corresponds to our target group (15)Application of domain-specific and domain-general training of math abilities (37, now 22) is analogy with our research design.

Thank you very much for all comments. They helped us to increase the scientific quality and "readability" of the text.

They were very encouraging.

Reviewer 3 Report

This paper describes a study that investigates whether a domain-specific in- tervention targeting maths and executive functions of primary school children with a Roma background would be effective in improving their scholastic performance and executive functioning. The study involved 122 4th-grade students. The study aims to investigate whether a newly developed intervention stimulating executive functioning, ExeFun-Mat (executive functioning stimulation via mathematics), would be beneficial in improving Roma children's executive functioning and scholastic performance in the field of maths. The results con- clude that both maths performance and executive functions improved over time in the class, with no significant differences between the domain-specific intervention group, active control group, and passive control group.

The introduction and theoretical framework sections are well-written and insightful. The area of study is intriguing and the background with a multitude of recent related references adds potency to the relevance of the paper. Especially motivating is the idea that In recent studies it was found that the all-important executive functioning can be strengthened as a consequence of intervention.

Specifically the intervention was the Exefun-Mat program, in which one trained administrator worked with two students. The intervention was based on the principles of the reciprocal teaching approach, scaffolding and self-questioning. This included (1) focusing on the student’s ability to generate questions, (2) clarifying and summarizing the information they have read, (3) moving from being passive observers of learning to active teachers, (4) becoming involved in the learning experience as peer tutors.

The research sample consisted of very low achievers in Math, many of them with (1) a history of a grade repetition, (2) low conduct of the language of instruction, and (3) observed low level of motivation. The findings indicated that in all groups of children, both for those who received the ExeFun-Mat program, and those in the control conditions, executive functions seemed to improve over time, but the experimental condition did not bring about significant improvement. The paper astutely observes that although the literature demonstrates a strong correlation between executive function and achievement, the two may not be causally related. This adds merit to the discussion section.

In conclusion, the current research found that the ExeFun-Mat program that was implemented among children from Roma backgrounds in grade 4 did not lead to substantial improvements in their executive functions, nor their math abilities, over and beyond those brought about by an alternative educational program.

The paper reports findings that are distinct from those reported in the pre- vious literature and elucidates reasoning to explain this. In particular the part about how the intervention in this study was provided relatively late, as the children already had significant delay in mathematical skills compared to their peers I found particularly relevant. And I agree wholeheartedly that devoting extra opportunities to teaching and learning may improve educational outcomes, but the timing is very important.

The area of research is exciting and potentially impactful. The presentation of thoughts in the paper is clear. The study design is sound and experimental results reporting is extensive.

Note: low-preforming students in the title should be low-performing students

Author Response

Response to Reviewer 3 Comments

  • Note: low-preforming studentsin the title should be low-performing students

Corrected.

Thank you very much for reading and commenting the text.  They helped us to increase the scientific quality and "readability" of the text.

They were very encouraging.
